# Synergistic Neuroprotective Effect of *Schisandra chinensis* and *Ribes fasciculatum* on Neuronal Cell Death and Scopolamine-Induced Cognitive Impairment in Rats

**DOI:** 10.3390/ijms20184517

**Published:** 2019-09-12

**Authors:** Eunkuk Park, Min Jeong Ryu, Nam Ki Kim, Mun Hyoung Bae, Youngha Seo, Jeonghyun Kim, Subin Yeo, Memoona Kanwal, Chun Whan Choi, Jun Young Heo, Seon-Yong Jeong

**Affiliations:** 1Department of Medical Genetics, Ajou University School of Medicine, Suwon 16499, Korea; 2Department of Biomedical Sciences, Ajou University Graduate School of Medicine, Suwon 16499, Korea; 3Department of Biochemistry, Chungnam National University School of Medicine, Daejeon 301747, Korea; 4Research Institute for Medical Science Chungnam National University School of Medicine, Daejeon 301747, Korea; 5Rpbio Research Institute, Rpbio Co. Ltd., Suwon 16229, Korea; 6NineB Research Institute, Nine B Co. Ltd., Suwon 16499, Korea; 7Natural Products Research Team, Gyeonggi Business & Science Accelerator, Suwon 16229, Korea; 8Department of Medical Science, Chungnam National University School of Medicine, Daejeon 301747, Korea; 9Infection Control Convergence Research Center, Chungnam National University School of Medicine, Daejeon 301747, Korea

**Keywords:** herbal medicine, *Schisandra chinensis*, *Ribes fasciculatum*, neuroprotection, neuronal cell death, ccopolamine-induced cognitive impairment

## Abstract

Mild cognitive impairment (MCI) is considered as a transitional stage between aging and Alzheimer’s disease. In the present study, we examined the protective effect of *Schisandra chinensis* (SC) and *Ribes fasciculatum* (RF) on neuronal cell death in vitro and scopolamine-induced cognitive impairment in Sprague Dawley^®^ rats in vivo. A mixture of SC and RF extracts (SC+RF) significantly protected against hydrogen peroxide-induced PC12 neuronal cell death. The neuroprotective effect of SC+RF on scopolamine-induced memory impairment in rats was evaluated using the passive avoidance test and the Morris water maze test. In the passive avoidance test, SC+RF-treated rats showed an increased latency to escape, compared to the scopolamine-treated rats. Moreover, SC+RF treatment significantly reduced escape latency in water maze test, compared to treatment with scopolamine alone. To verify the long-term memory, we performed probe test of water maze test. As a result, rat treated with SC+RF spent more time in the target quadrant. Consistent with enhancement of memory function, the brain derived neurotrophic factor (BDNF) and its downstream molecules (pERK, pATK, and pCREB) are increased in SC+RF treatment in hippocampal area compared with scopolamine treated group. These results suggest that a mixture of SC and RF extracts may be a good therapeutic candidate for preventing mild cognitive impairment.

## 1. Introduction

Mild cognitive impairment (MCI) is known as early dementia and isolated memory impairment, which commonly affects elderly people [1]. Delays or prevention of MCI are important because people with MCI are at high-risk for developing Alzheimer’s or other dementia [2,3]. However, United States Food and Drug Administration (FDA)-approved drugs inhibiting the cholinesterase or *N*-methyl-d-aspartate receptor, such as Aricept (donepezil), Exelon (rivastigmine), and Namenda (memantine), only mitigate the symptoms of Alzheimer’s disease [4,5,6]. Moreover, some of these drugs should not be used in the long-term due to how they affect heart rate and the nervous system, causing side effects such as arrhythmia [7]. To overcome the drawback of modern therapeutics, natural plants have emerged as alternative medicine in prevention therapy for transitional state of disease, which have low-risk of developing side effects [8].

*Schisandra chinensis* (SC), a deciduous woody vine, has been used as a traditional herbal medicine for the treatment of cancer and diabetes via immunomodulatory and antioxidant activities [9]. Bioactive compounds such as schisandrin B, a dibenzocyclooctadiene derivative isolated from SC, have been studied for their neuroprotective effects in scopolamine-induced amnesia, amyloid beta (Aβ) peptide-induced neurotoxicity, and cisplatin-induced neurotoxicity [10,11]. In addition, schisanhenol isolated from SC has been shown to improve learning and memory in scopolamine-treated mice [12]. Although protective effects of various extracts of SC were reported on neuronal injury and memory impairment, the underlying mechanism of it still did not defined with the brain derived neurotrophic factor (BDNF) signaling, and revealed that correlation with antioxidant enzyme activation [13,14], NF-kB inhibition [15], and recovered Ach levels [10,16].

Here, we found out protective role of neuronal cell death by *Ribes fasciculatum* (RF), a perennial woody deciduous member of the *Ribes* genus in the family Grossulariaceae, is distributed widely in Korea and China. Previous studies have reported that RF has been shown to increase the lifespan and stress resistance of *Caenorhabditis elegans* through SIR-2.1-mediated DAF-16 activation depending on the insulin/insulin growth factor signaling pathway [17]. Moreover, RF inhibited the transcription of nuclear factor of activated T cells (NFAT) [18] and the secretion of inflammatory cytokines in a mouse model of atopic dermatitis [19]. Since nuclear factors of activated T cells (NFATc4) have been reported to be required for BDNF-dependent survival of adult-born neurons and spatial memory formation in the hippocampus, we focused on the possibility that combination of SC and RF could enhance protection against cognitive impairment through the BDNF signaling pathway.

Ultimately, in the present study, we aimed to investigate whether SC and RF extracts have a protective effect on reactive oxygen species (ROS)-induced cytotoxicity in pheochromocytoma PC12 cells and scopolamine-induced cognitive impairment in rats.

## 2. Results

### 2.1. Verification of Increasing Hydrogen Peroxide (H_2_O_2_) Concentrations on Neuronal Cell Death

Neuronal apoptosis in neurodegenerative diseases is associated with ROS production. Therefore, we tested the effect of H_2_O_2_ on neuronal cell death in the PC12 neuronal cell line. Cells were treated with various concentrations of H_2_O_2_ (0–100 μM) for 1 h and neuronal apoptosis was measured by the WST assay. Neuronal cell death occurred in a concentration-dependent manner. A 1 h incubation with 50 μM H_2_O_2_-induced approximately 40% of the cells to undergo apoptosis; finally, we selected 50 μM H_2_O_2_ for the subsequent experiments (Figure 1).

### 2.2. Protective Effect of SC and RF on H_2_O_2_-Induced Neuronal Cell Death

To identify the effectiveness of herbal medicine against nerve cell loss, we screened 42 herbal medicines with PC12 cells. Further, we selected SC and RF extracts due to have higher potency than other candidates (Appendix A). To investigate the protective effect of ethanol extracts of SC and RF against H_2_O_2_-induced neuronal cell death, we incubated PC12 neuronal cells for 24 h with either of three different concentrations of SC and RF (2, 10, or 50 μg/mL) extracted at various ethanol percentages (0%, 30%, 70%, or 100%). Cells were pre-incubated with SC and RF for 23 h and then co-incubated with hydrogen peroxide for 1 h. In previous reports, SC extracted with 50% ethanol was most effective when performing various chemical properties such as total phenolic compounds (TPC) and antioxidant activity (2,2-diphenyl-1-picrylhydrazyl (DPPH) and 2,2’-azino-bis 3-ethylbenzothiazoline-6-sulphonic acid (ABTS) analysis) [20]. Consistent with the previous report, cells treated with 30% ethanol extracts of SC and RF had the highest survival rate, compared to those treated with 0%, 70%, or 100% ethanol extracts. The greatest neuroprotective effect was observed at a concentration of 50 μg/mL for both 30% ethanol extracts of SC and RF (Appendix A and Figure 2). However, SC and RF did not have cytotoxic effects (Appendix A). These results suggested that the 30% ethanol extracts of SC and RF protected against H_2_O_2_-induced reduction of cell viability without exerting a cytotoxic effect.

To investigate a synergistic effect of SC and RF extracts against H_2_O_2_-induced cell death, we determined the cell survival rate of PC12 neuronal cells treated with various ratios of SC and RF (6:4, 7:3, or 8:2). All SC+RF combinations inhibited H_2_O_2_-induced cell death. The greatest protective effect against H_2_O_2_-induced cell death occurred when a 7:3 ratio of 50 μg/mL SC+RF was administrated (Figure 3A). The mixture did not have any cytotoxic effects (Appendix A). Interestingly, the mixture of 50 μg/mL SC+RF exerted a greater protective effect than the single extracts of SC or RF at 50 μg/mL (Figure 3B). These results suggested that 30% ethanol extracts of SC and RF at a 7:3 ratio exerted a synergistic protective effect against H_2_O_2_-induced cell death in PC12 neuronal cells. Next, we analyzed the main constituent of single extract of SC and RF and combination of SC and RF (7:3), using a high-performance liquid chromatography-electrospray ionization-tandem mass spectrometry (HPLC-ESI-MS) system. Ion current chromatogram of the SC and RF and combination of SC and RF (7:3) showed the most abundant constituents identified in SC (shizandrin) and in RF (4-hydrobenzoic acid) (Appendix A). Although we did not perform HPLC with the content of the extracted compounds under all extraction conditions, pharmacological studies have been conducted on active extracts that have a protective effect on hydrogen peroxide-induced neuronal cell death. Taken together, we assert that shizandrin and 4-hydrobenzoic acid are major constituents with 30% ethanol extract that have a protective effect on hydrogen peroxide-induced neuronal cell death.

### 2.3. A Mixture of SC and RF Extracts Prevent Scopolamine-Induced Cognitive Impairment in Rats

Based on the in vitro study, we further investigated the neuroprotective effect of an extract mixture of SC and RF (SR) on scopolamine-induced memory impairment in vivo. Scopolamine is a drug that can induce cognitive deficits in healthy humans and animals [21]. The neuroprotective effect of SC and RF on scopolamine-induced memory impairment in rats was evaluated by the passive avoidance test and Morris water maze test. The rats used were seven week old Sprague Dawley^®^ males and were orally administered a mixture of SC+RF (75, 150, or 300 mg/kg/day) for 23 days before a scopolamine injection (1 mg/kg, injected intraperitoneally). The passive avoidance test and Morris water maze test were performed 30 min after scopolamine administration. Phosphatidylserine (PS), a phospholipid that reduces the risk of dementia and cognitive dysfunction in the elderly, as announced by the FDA, was used as a positive control. As expected, PS treatment restored scopolamine-induced spatial learning and memory impairment. Comparing with PS treatment, in passive avoidance test, rats treated with SC+RF showed increased latency time (Figure 4A), without changes in body weight (Appendix A). This result suggests that SC and RF improve memory based on fear motivation. In the Morris water maze test, we did not find significant differences between rats treated with scopolamine and SC+RF (SR) and rats treated with scopolamine alone until day four (Figure 4B and Appendix A). However, at day five, rats treated with scopolamine and SR (75 or 150 mg/kg/day) showed significantly reduced escape latency value, compared to scopolamine-treated rats (Figure 4B,C), without changes in body weight (Appendix A). On the water maze test, learning and memory were assessed on a probe trial. Rats treated with SC+RF spent more time in the target quadrant (Figure 4D). This result indicates that treatment of SC and RF restored scopolamine-induced spatial learning and memory impairment.

### 2.4. A Mixture of SC and RF Extracts Exerts Neuroprotection via BDNF Signaling in Hippocampus

BDNF have function in differentiation and survival of neurons of the CNS and in the enhancement of synaptogenesis and synaptic plasticity in the adult CNS [22,23]. To verify the change of BDNF using SC and RF extracts, we performed the western blot analysis. Consistent with behavioral test, BDNF was inhibited by scopolamine, and recovered by treatment with PS, but increased further by administration of SC and RF (Figure 5A,B). To identify the involvement of BDNF signaling pathway by SC and RF in scopolamine treated rat model, we measured the phosphorylation of extracellular signal-regulated kinase (ERK), cAMP response element binding (CREB) protein, and AKT/protein kinase B (PKB). Surprisingly, treatment of PS did not restore scopolamine-induced reduction of phosphorylated ERK, CREB and AKT (Figure 5A,C–E).

Taken together, these results indicate that a mixture of SC and RF extracts prevents scopolamine-induced long-term spatial learning and memory impairment, through activation of BDNF-induced signaling cascade.

## 3. Discussion

Alzheimer’s disease is an age-related neurodegenerative disorder characterized clinically by the progressive degeneration of memory and cognitive functions by hippocampal neuronal loss [24,25]. In this study, we investigated the neuroprotective effect of SC and RF on oxidative stress induced by H_2_O_2_ in a neuronal cell line in vitro and on scopolamine-induced memory impairment in rats in vivo.

Oxidative stress and inflammation have been implicated in neurodegeneration, including cognitive impairment [26]. SC has been extensively studied for its immunomodulatory, antioxidant, anticancer, and cognition/memory-improving activities [10,11,27,28]. In addition, SC has been shown to protect the liver, kidney, and nervous system in cyclophosphamide-treated rats [27]. Studies have reported that SC and its active ingredients protect against neurological diseases [29,30] and improve cognitive impairment in Aβ 1-42-induced neurodegeneration in mice [31]. In turn, RF inhibits the NFAT transcription factor [17] and exerts a pharmacological action in allergic inflammatory diseases through the reduction of nuclear factor-kappa B activation in lipopolysaccharide-stimulated macrophages [32]. Recently, RF has been shown to have anti-aging properties via increased longevity and stress resistance in *C. elegans* [19]. In the present study, both SC and RF prevented neuronal cell death were induced using H_2_O_2_. Among a range of ethanol extracts, the 30% ethanol extracts of SC and RF were the most effective in reducing neuronal cell death. Furthermore, a synergistic effect of 30% ethanol extracts of SC+RF was observed at a 7:3 combination ratio. Compared to the single extracts of SC or RF, a mixture of SC+RF showed greater protection against neuronal cell death. These results suggest that a combination of SC and RF prevented oxidative stress-induced neuronal cell death without undesirable adverse effects.

Scopolamine has been used to generate a model of amnesia in mice that leads to increased oxidative stress with impairment of memory and cognitive functions [33]. The amnesic effect of scopolamine has also been demonstrated in rodents [34]. Based on the results of our in vitro experiment, we investigated the neuroprotective effect of SC and RF on scopolamine-induced memory impairment in rats. Cognitive impairment was evaluated by the Morris water maze and the passive avoidance test. The Morris water maze has been extensively used in the behavioral neurosciences to assess spatial learning and memory [35]. Since rats in each group entered the dark compartment immediately after being placed in the illuminated compartment, the latency time during the training period was very short. The passive avoidance test is a fear-aggravated test used to evaluate learning and memory. We found that scopolamine-treated rats showed increased escape latency and decreased time in the target quadrant, compared to the control group. In contrast, treatment with SC+RF increased escape latency and time in the target quadrant in scopolamine-treated rats. Similarly, scopolamine disrupted passive avoidance memory retrieval, but the latency time was improved by treatment with SC+RF. These results suggest that SC and RF may prevent scopolamine-induced memory impairment in rats.

Regarding the enhancement of cognitive and memory function, we assure that the BDNF signaling are associated with neuroprotective effect of SC and RF treatment. BDNF is a predominant neurotrophic factor involved in AD that activates Tropomyosin receptor kinase B (TrkB) receptors and phosphorylates signaling proteins to induce neural differentiation and survival [36]. Recently, numerous reports suggested that BDNF can affect synaptogenesis and synaptic plasticity in adult central nervous system (CNS) [22,37,38,39]. Of note, SC could modulate BDNF, TrkB/ cAMP response element binding protein (CREB)/ extracellular-signal-regulated kinase (ERK), and Phosphoinositide 3-kinase (PI3K)/AKT/glycogen synthase kinase 3 beta(GSK-3β) in the hippocampus and attenuate the depression-like emotional status and associated cognitive deficits in chronic unpredictable mild stress mice or corticosterone-induced mice [40,41]. In our scopolamine-induced cognitive impairment model, SC and RF also ameliorated with activation of BDNF and its downstream signaling pathway (CREB/ERK/AKT). By activation of the BDNF, neurotrophic signaling stimulated mainly through the activation of CREB/ERK/AKT pathway which ultimately improved neuronal cell survival, differentiation, growth, synaptic plasticity, and long-term memory. Thus, activation of BDNF-induced signaling cascade by treatment of SC and RF mediated the neuroprotective effects against scopolamine-induced animal model. Underlying mechanism of combined therapy, specifically RF treatment, for protective effects and synaptic plasticity in scopolamine induced rats is needed further investigation.

## 4. Materials and Methods

### 4.1. Preparation of SC and RF Extracts

A total of 42 plant extracts were originally provided from the Korea Plant Extract Bank at Korea Research Institute of Bioscience and Biotechnology (KRIBB) (Ochang, Chungbuk, Korea) [42]. The ethanol extracts were prepared based on a previous report [43]. Briefly, after botanical authentication, the dried plant material was turned into powder using a grinding mill. Next, 100 g of plant powder was extracted in 1000 mL of 0%, 30%, 50%, 70%, and 100% aqueous ethanol solution (v/v) in an occasional shaker at room temperature for 72 h. The extract was then filtered with Whatman No. 2 filter paper (GE Healthcare Life Sciences, Logan, UT, USA) to remove debris, and the filtrate was collected and concentrated in a rotating evaporator at 40–50 °C and lyophilized under reduced pressure. The dried extract was stored in amber flasks at 4 °C, re-suspended with distilled water, and filtered through a 0.22-μm membrane filter (Sartorius, Gottingen, Germany) before the experiments.

### 4.2. Cell Culture and Cytotoxicity Assay

The rat PC12 cell line was purchased from the Korean Cell Line Bank (Seoul, Korea) and cultured at 37 °C and 5% CO_2_ with Dulbecco’s Modified Eagle Medium (Thermo Fisher Scientific, Inc., Waltham, MA, USA) supplemented with 10% fetal bovine serum (GE Healthcare Life Sciences), 100 U/mL penicillin, and 100 mg/mL streptomycin. The PC12 cells (1 × 10 ^4^ cells per well) were plated in 96-well plates. To test the effect of H_2_O_2_ on neuronal cell death in the PC12 neuronal cell line, cells were treated with various concentrations of H_2_O_2_ (Sigma Aldrich, Inc., St. Louis, MO, USA) (0–100 μM) for 1 h and WST assay was performed. To test the protective effect of SC and RF in H_2_O_2_ induced neuronal cell death, the cells were pre-incubated with different concentrations of SC and RF (2, 10, or 50 μg/mL) for 23 h and then added with 50 μM of H_2_O_2_ for 1 h. We used the EZ-Cytox Cell Viability Assay Kit (Daeil, Korea) to assess cell viability. After serial treatment of SC, RF, and hydrogen peroxide, WST (20 μL; 5 mg/mL in phosphate-buffered solution) was added to each well and the cells were incubated for another 4 h. WST is reduced by dehydrogenase in cells to form an orange-colored formazan. The amount of formazan and the number of living cells are in direct proportion. Absorbance was measured at wavelength of 450 nm and 655 nm (for reference) using a microplate reader (Bio-Rad, Hercules, CA, USA)

### 4.3. In Vivo Experiments in Scopolamine-Treated Rats

In this study, seven-week-old male Sprague Dawley^®^ rats were used. Animals were allocated in groups and were allowed to accommodate for one week in the animal house. Rats were provided with a standard chow diet and were given water ad libitum. During the experimental period, the animals were kept at a temperature of 22 ± 3 °C, a 12-h light/dark cycle, and a relative humidity varied between 40% and 60%. Animals were treated with either of six different conditions: (1) vehicle; (2) scopolamine (1 mg/kg) (Sigma Aldrich, Inc.); (3) scopolamine with 50 mg/kg/day of PS as a positive control; (4) scopolamine with 75 mg/kg/day of SC+RF (SR (75)); (5) scopolamine with 150 mg/kg/day of SC+RF [SR (150)]; and (6) scopolamine with 300 mg/kg/day of SC+RF (SR (300)). Rats were orally administrated SC+RF for 28 days. The animal research protocol was approved by the Animal Care and Use Committee of the Ajou University School of Medicine, and all experiments were conducted in accordance with the institutional guidelines established by the Committee.

### 4.4. Behavioral Studies

#### 4.4.1. Passive Avoidance Test

A passive avoidance test was conducted as described previously [44] with mild modification. The device used for the passive avoidance test consisted of an illuminated and a dark room separated by an acrylic plate featuring small passages (total volume, 490 × 250 × 300 mm^3^; Daejong Inc., Korea). After 23 days of treatment with SC+RF, the learning phase was performed for five days. The rats were placed in the illuminated section without light and adaptation was allowed for 60 s. Then, when the light was turned on and the guillotine door was opened, the rat entered the dark compartment through the open passage. After entering the dark compartment, a foot shock (0.5 mA for 3 s) was delivered to the floor grid in the dark compartment as the guillotine door closed. After 24 h of the training trial, the acquisition test was carried out 30 min after an intraperitoneal injection of 1 mg/kg scopolamine. The time spent in the illuminated compartment was defined as the latency period. The test was terminated if the rats did not enter the dark room for 300 s.

#### 4.4.2. Morris Water Maze

The Morris water maze test was carried out as described previously [45] with slight modification. The water maze consisted of a large circle pool and divided into four equal quadrants. A white plastic platform was submerged in one of the quadrants of the pool, 1 cm below the surface of the water. A large circle pool was filled to a depth of 30 cm with opaque water containing kids’ paint (Black Kidsmomart, Korea) so as to hide the plastic platform. The rats were allowed to stay on the white plastic platform for 10 s and the experiment was started four times in different directions. After 23 days of treatment with SC+RF, the Morris water maze test was performed for five days. A SMART video tracking system (Smart 3.0; Panlab Harvard Apparatus, Barcelona, Spain) was used to monitor and analyze all swimming activity and to record escape latency, swim distance, and speed. The test was carried out 30 min after an intraperitoneal injection of 1 mg/kg scopolamine (Sigma Aldrich, Inc.). The time taken to find the hidden platform was recorded as the escape latency.

### 4.5. Protein Isolation and Western Blotting

Proteins of hippocampus tissues were extracted using RIPA buffer (1% Nonidet P-40, 0.1% SDS, 150 mM NaCl, 50 mM Tris–HCl pH 7.5, and 0.5% deoxycholate) with 10% of phosphatase inhibitor and protease inhibitor (Roche, Basel, Switzerland). Proteins were loaded on SDS-PAGE gel and run by electrophoresis, and then transferred to polyvinylidene fluoride (PVDF) membrane. Membrane blocked by 5% skim milk for 1 h, and incubated with primary antibody including anti-pERK (9102, Cell signaling Technology, Beverly, MA, USA), anti-tERK (9101s, Cell signaling Technology), anti-BDNF (ab108319, Cambridge, MA, USA), anti-pCREB (9198, Cell signaling Technology), anti-tCREB (9197, Cell signaling Technology), anti-pAKT(9271, Cell signaling Technology), anti-tAKT (9272s, Cell signaling Technology), and anti-Actin (sc-1616, Santa Cruz, CA, USA) antibodies at 4 °C for overnight. Anti-IgG horseradish peroxidase antibody (Pierce Biotechnology, MA, USA) correspond with the host of primary antibody was used as secondary antibody. Proteins band was detected by ECL system (Thermo Scientific, Inc.). The quantification of band intensity was normalized with actin using Image J program (version 1.52c; National Institutes of Health, Bethesda, MD, USA).

### 4.6. Statistical Analysis

A statistical software package (SPSS 11.0 for Windows, SPSS Inc., Chicago, IL, USA) was used to perform the statistical tests. The statistical significance of the differences between groups was assessed by the Student’s *t*-test. Comparisons of multiple groups were done with a one-way analysis of variance, followed by Tukey’s honest significant difference post-hoc test for correction of multiple comparisons. A probability value (*p*) less than 0.05 was considered statistically significant. The results are expressed as mean ± standard error of the mean and all experiments were repeated at least three times.

## 5. Conclusions

Our study shows that SC and RF protected neuronal cells against H_2_O_2_-induced cell death. Moreover, SC and RF prevented cognitive impairment via BDNF signaling pathway in a scopolamine-induced rat model. Our findings indicate the potential of SC and RF for use in the prevention and treatment of cognitive impairment in humans with Alzheimer’s disease.

## Figures and Tables

**Figure 1 ijms-20-04517-f001:**
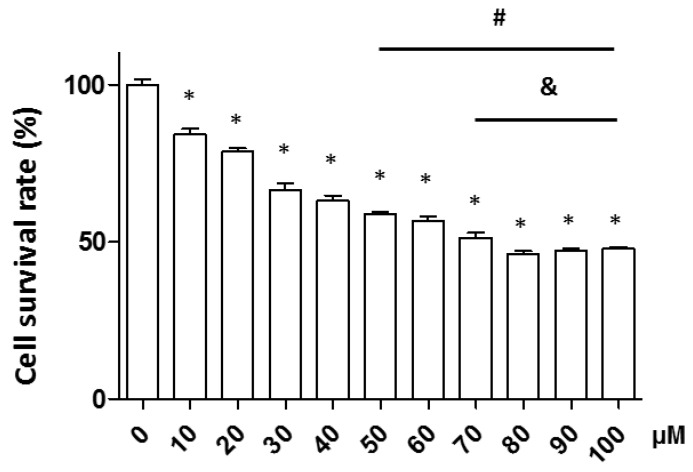
Neuronal cell survival rate in various concentrations of H_2_O_2_. Cells were treated with various concentrations of H_2_O_2_ (0, 10, 20, 30, 40, 50, 60, 70, 80, 90, or 100 µM) and cell viability was measured by the tetrazolium salt (WST) assay. The data shown are means ± standard error of the mean (SEM). *: *p* < 0.05 vs. 0 µM, #: *p* < 0.05 vs. 30 µM, and &: *p* < 0.05 vs. 40 µM.

**Figure 2 ijms-20-04517-f002:**
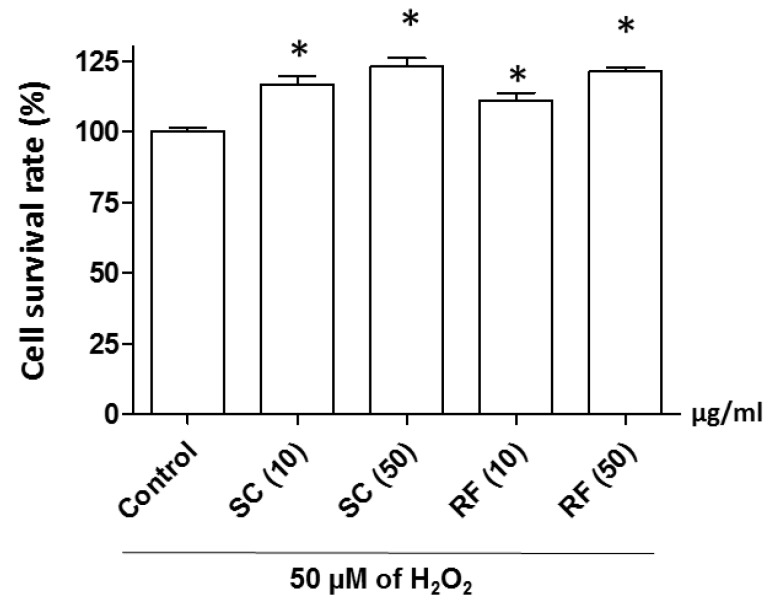
Effect of *Schisandra chinensis* (SC) and *Ribes fasciculatum* (RF) extracts on cell survival rate. SC and RF were extracted with 30% ethanol and PC12 neuronal cells were treated with SC and RF extracts at 10 or 50 μg/mL. The protective effect of SC and RF extracts were measured by the WST assay. Control is a treatment of 50 µM of H_2_O_2_. *: *p* < 0.05 vs. control.

**Figure 3 ijms-20-04517-f003:**
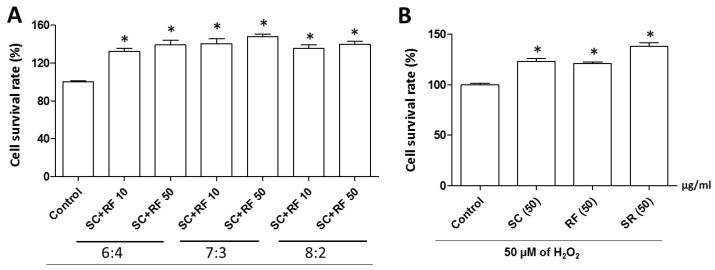
Effect of single or combined *Schisandra chinensis* (SC) and *Ribes fasciculatum* (RF) extracts on neuronal cell survival rate. PC12 neuronal cells were treated with different ratios (6:4, 7:3, or 8:2) of SC and RF at concentrations of 10 or 50 μg/mL (**A**). The effect of single or combined SC and RF extracts (SR) on cell survival rate was compared (**B**). Control is a treatment of 50 µM of H_2_O_2_. *: *p* < 0.05 vs. control.

**Figure 4 ijms-20-04517-f004:**
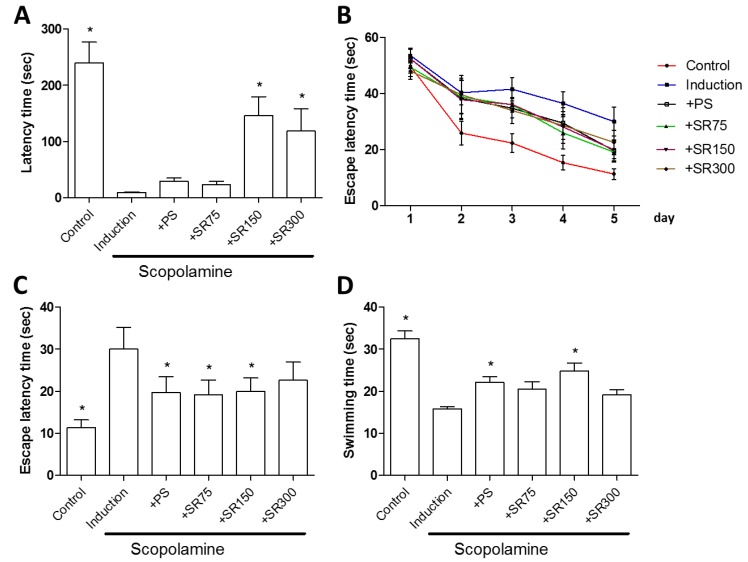
Effect of SC and RF extracts on scopolamine-induced cognitive impairment in scopolamine-treated rats. Rats were treated orally with SC+RF (SR) (75, 150, or 300 mg/kg/day) for 23 days and memory impairment was measured by the passive avoidance test and Morris water test. Fear motivation in the passive avoidance test was evaluated by latency time in scopolamine-treated rats (**A**). Escape latency time was measured daily for five days (**B**) and a significantly reduced escape latency time in SR treated-rats (75 or 150 mg/kg/day) was observed at day five (**C**). On the last day of the Morris water maze, swimming time in the target quadrant was measured (**D**). *: *p* < 0.05 vs. Induction. Abbreviation: PS, phosphatidylserine.

**Figure 5 ijms-20-04517-f005:**
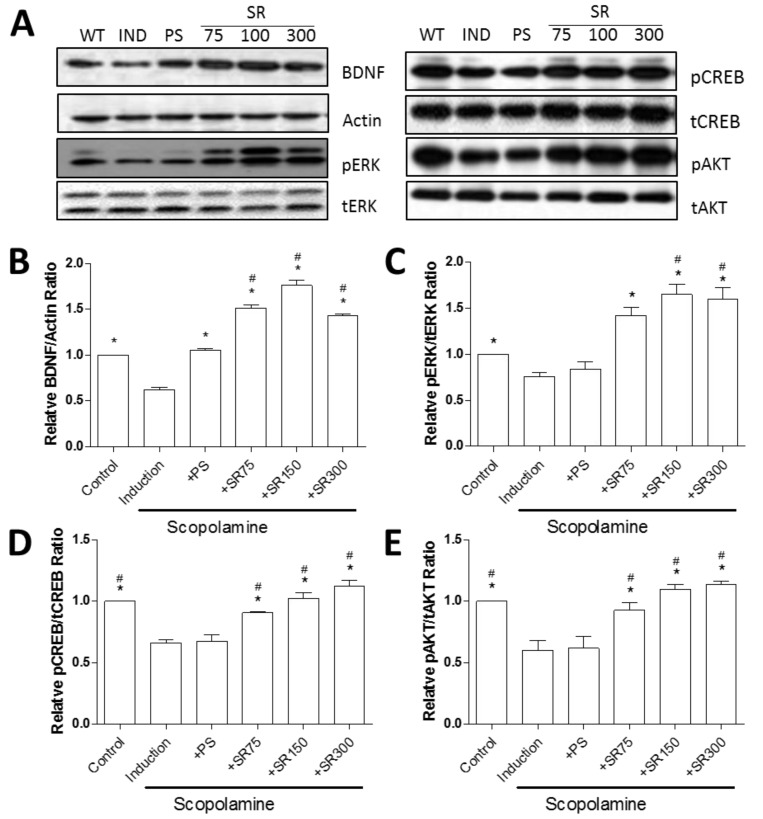
Involvement on brain derived neurotrophic factor (BDNF) signaling pathway for treatment of SC and RF extracts. Rats were treated orally with SC+RF (SR) (75, 150, or 300 mg/kg/day) for 28 days. Hippocampus was extract 30 min after an intraperitoneal injection of 1 mg/kg scopolamine. Western blot analysis with specific antibodies and Actin was used for loading control (**A**). Relative band ratio was calculated with image J (**B**–**E**). *: *p* < 0.05 vs. Scopolamine induction. #: *p* < 0.05 vs. PS. Abbreviation: PS, phosphatidylserine. SCO, Scopolamine.

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
