# Peer review of "Synergistic Neuroprotective Effect of Schisandra chinensis and Ribes fasciculatum on Neuronal Cell Death and Scopolamine-Induced Cognitive Impairment in Rats"

_ijms, 2019, doi:10.3390/ijms20184517_

Round 1
Reviewer 1 Report
The authors describe the neuroprotective effects of extracts of Schisandra chinensis and Ribes fasciculatum using in vitro (PC12 cells) and in vivo (using rats) models. The results are very interesting and clearly demonstrate the beneficial effects of the extracts, and deserve to be published. But, first the authors need to correct and clarify some point of the manuscript.
Title. The authors should write the species name adequately (Schisandra chinensis; Ribes fasciculatum) in italic
Introduction:
Line 60. authors should define “Aβ-“
Results:
As some Extracts HPLC analysis is shown in supplementary material, authors should mention the major compounds in extracts, and the differences between extracts, i.e., which are the major differences in extracts composition extracted with different % of ethanol. Or, if this information is published in other article mention it in introduction.
Line 74-80, subsection 2.1. In this section it is not clear is the cells exposure to hydrogen peroxide is in the presence or absence of extracts. It seems to be in the absence, but in the methods section authors mention that incubation of cells with hydrogen peroxide is preceded by extracts incubation. Please make sure that these points are clear to a better understanding of results.
In line 242 (methods) authors mention 24 h incubation. In line 85 authors mention 48 h incubation. Please clear this point. Line 87 (results) does not correlates with the method described in section 4.1. Preparation of SC and RF Extracts. In methods authors only mention extraction with 70% ethanol. In results sections it seems that various types of extracts were performed.The method of extraction, as well as the extracts used should be clearly described. Line 87: Please indicate if the medium containing the extracts was removed prior to hydrogen peroxide incubation, or if it was co-incubations.
Figure 2. Authors should mention what is control. Is control non-treated cells? If so the extracts induce cell growth? Or is the control cells treated with hydrogen peroxide in the absence of extracts? This figure should have the control (without extracts and without hydrogen peroxide) in order to see the beneficial effect of extracts. Figure 3. according to the text description, it seems that the authors did not place correctly the legends of 7:3 (SC+RF 10 vs 50). Please verify. (also mention in legend what is control).
Line 129. Phosphatidylserine (PS), is a phospholipid (should be designated as phospholipid)
Discussion. Is ok, but could be improved with more recent articles, such as:
Nutrients 2019, 11, 1671; doi:10.3390/nu11071671 (Neuroprotective Efect of Schisandra Chinensis on Methyl-4-Phenyl-1,2,3,6-Tetrahydropyridine-Induced Parkinsonian Syndrome in C57BL/6 Mice)
And others…
Methods: are adequately described, only a few minor points, as follows:
L.233. “and the supernatant was”, I think the authors wanted to mean the filtrate instead of supernatant
L.242-243: The authors should mention if the H2O2 treatment was performed in the presence of extracts or if the extracts incubation media was removed prior to challenge the cells with hydrogen peroxide.
Also the concentration of hydrogen peroxide should be mentioned (the range of concentrations).
L245-246. Please verify if there is a missing step.
Supplementary material. Supplementary Table 1., should mention in its legend the type of extracts (% of ethanol).
Author Response
Comments and Suggestions for Authors
The authors describe the neuroprotective effects of extracts of Schisandra chinensis and Ribes fasciculatum using in vitro (PC12 cells) and in vivo (using rats) models. The results are very interesting and clearly demonstrate the beneficial effects of the extracts, and deserve to be published. But, first the authors need to correct and clarify some point of the manuscript.
We deeply thank the reviewer for providing many constructive comments, and have addressed each point in the appended rebuttal document. We made clear mention of the treatment of hydrogen peroxide in each experiment and the extraction of Schisandra chinensis and Ribes fasciculatum.
Q1) Title: The authors should write the species name adequately (Schisandra chinensis; Ribes fasciculatum) in italic.
A1) Thank you for comment. We change the species name adequately (Schisandra chinensis; Ribes fasciculatum) in italic.
Q2) Introduction: Line 60. authors should define “Aβ-“
A2) Thank you. We define the amyloid beta (Aβ) peptide on line 18, page 3, as shown below.
“Bioactive compounds such as schisandrin B, a dibenzocyclooctadiene derivative isolated from SC, have been studied for their neuroprotective effects in scopolamine-induced amnesia, amyloid beta (Aβ) peptide-induced neurotoxicity, and cisplatin-induced neurotoxicity.”
Q3) Results: As some Extracts HPLC analysis is shown in supplementary material, authors should mention the major compounds in extracts, and the differences between extracts, i.e., which are the major differences in extracts composition extracted with different % of ethanol. Or, if this information is published in other article mention it in introduction.
A3) Thank you very much for your valuable comments. To identify the protective effect of ethanol extracts of SC and RF against H2O2-induced neuronal cell death, we performed cell viability assay with SC and RF (2, 10, or 50 μg/ml) extracted at various ethanol percentages (0%, 30%, 70%, or 100%), followed by 1 h incubation of H2O2 in PC12 neuronal cells. In previous reports, SC extracted with 50% ethanol was most effective when performing various chemical properties such as total phenolic compounds (TPC) and antioxidant activity (2,2-diphenyl-1-picrylhydrazyl(DPPH) and 2,2'-azino-bis 3-ethylbenzothiazoline-6-sulphonic acid (ABTS) analysis)[1]. Consistence with previous report, the cells treated with 30% ethanol extracts of SC and RF had the highest survival rate, compared to those treated with 0%, 70%, or 100% ethanol extracts. Next, to investigate a synergistic effect of SC and RF extracts against H2O2-induced cell death, we determined the cell survival rate of PC12 neuronal cells treated with various ratios of SC and RF (6:4, 7:3, or 8:2). As a results, 30% ethanol extracts of SC and RF at a 7:3 ratio exerted a synergistic protective effect against H2O2-induced cell death in PC12 neuronal cells. To know the main constituent of single extract of SC and RF and combination of SC and RF (7:3), we performed high-performance liquid chromatography-electrospray ionization-tandem mass spectrometry (HPLC-ESI-MS). Ion current chromatogram of the SC and RF and combination of SC and RF (7:3) showed the most abundant constituents identified in SC (shizandrin) and in RF (4-hydrobenzoic acid). Although we did not perform HPLC with the content of the extracted compounds under all extraction conditions, pharmacological studies have been conducted on active extracts that have a protective effect on hydrogen peroxide-induced neuronal cell death. Taken together, we assert that shizandrin and 4-hydrobenzoic acid are major constituents with 30% ethanol extract that have a protective effect on hydrogen peroxide induced neuronal cell death. We add this description on line 4, page 6 and line 13, page 7.
Q4) Line 74-80, subsection 2.1. In this section it is not clear is the cells exposure to hydrogen peroxide is in the presence or absence of extracts. It seems to be in the absence, but in the methods section authors mention that incubation of cells with hydrogen peroxide is preceded by extracts incubation. Please make sure that these points are clear to a better understanding of results.
A4) Thank you for your comment. Subsection 2.1 is the study of neuronal cell death by various concentration of hydrogen peroxide alone. One-hour incubation with 50 μM hydrogen peroxide induce about 40% neuronal cell death. As a result, we proceed the subsequent experiments at 50 μM concentration.
To avoid confusion, we have clearly explained the treatment of hydrogen peroxide in the Methods section, on line18, page 14, as shown below.
“To test the effect of H2O2 on neuronal cell death in the PC12 neuronal cell line, cells were treated with various concentrations of H2O2 (0–100 μM) for 1 h and WST assay was performed. To test the protective effect of SC and RF in H2O2 induced neuronal cell death, the cells were pre-incubated with different concentrations of SC and RF (2, 10, or 50 μg/ml) for 24 h and then treated with H2O2 for 1 h.”
Q5) In line 242 (methods) authors mention 24 h incubation. In line 85 authors mention 48 h incubation. Please clear this point. Line 87 (results) does not correlates with the method described in section 4.1. Preparation of SC and RF Extracts. In methods authors only mention extraction with 70% ethanol. In results sections it seems that various types of extracts were performed. The method of extraction, as well as the extracts used should be clearly described. Line 87: Please indicate if the medium containing the extracts was removed prior to hydrogen peroxide incubation, or if it was co-incubations.
A5) Thank you for kind comment.
First, in all experiment, SC and RF were incubated for 24h in PC12 cells. In the Results section, we modified 48 h to 24 h.
Second, we re-wrote the percentage of EtOH in the Preparation of SC and RF Extracts in the Method section as shown below.
“Next, 100 g of plant powder was extracted in 1000 ml of 0%, 30%, 50%, 70% and 100% aqueous ethanol solution (v/v) in an occasional shaker at room temperature for 72 h”
Third, to investigate the protective effect of ethanol extracts of SC and RF against H2O2-induced neuronal cell death, we incubated PC12 neuronal cells for 24 h with either of three different concentrations of SC and RF (2, 10, or 50 μg/ml) extracted at various ethanol percentages (0%, 30%, 70%, or 100%). Cells were pre-incubate with SC and RF for 23 h and then co-incubated with hydrogen peroxide for 1 h. We added above description in the Result section on line 3, page 6.
Q6) Figure 2. Authors should mention what is control. Is control non-treated cells? If so the extracts induce cell growth? Or is the control cells treated with hydrogen peroxide in the absence of extracts? This figure should have the control (without extracts and without hydrogen peroxide) in order to see the beneficial effect of extracts. Figure 3. according to the text description, it seems that the authors did not place correctly the legends of 7:3 (SC+RF 10 vs 50). Please verify. (also mention in legend what is control).
A6) Thank you for constructive comment. As showed in supplementary figure 2 and 3, treatment of each SC and RF or a mixture of SC and RF did not enhance neuronal cell proliferation. However, each extract and mixture of extract have protective effects against hydrogen peroxide induced neuronal cell death (Figure 2, 3 and supplementary figure 1). We clearly mentioned in the result section that the control means 50 μM of H2O2 in Figure 2, Figure 3 and supplementary figure 1.
To investigate a synergistic effect of SC and RF extracts against H2O2-induced cell death, we determined the cell survival rate of PC12 neuronal cells treated with various ratios of SC and RF (6:4, 7:3, or 8:2). All SC+RF combinations inhibited H2O2-induced cell death. The greatest protective effect against H2O2-induced cell death occurred when a 7:3 ratio of 50 μg/ml SC+RF was administrated (Fig. 3A). And we have mistaken in input the labeling, so we modified the figure as below description.
Q7) Line 129. Phosphatidylserine (PS), is a phospholipid (should be designated as phospholipid)
A7) Thanks. We revised phosphatidylserine (PS) with phospholipid, not chemical.
Q8) Discussion. Is ok, but could be improved with more recent articles, such as:
Nutrients 2019, 11, 1671; doi:10.3390/nu11071671 (Neuroprotective Efect of Schisandra Chinensis on Methyl-4-Phenyl-1,2,3,6-Tetrahydropyridine-Induced Parkinsonian Syndrome in C57BL/6 Mice)
And others…
A8) Thank you. We added the reference with recent article.
“SC has been extensively studied for its immunomodulatory, antioxidant, anticancer, and cognition/memory-improving activities [2-5]. In addition, SC has been shown to protect the liver, kidney, and nervous system in cyclophosphamide-treated rats [4]. Studies have reported that SC and its active ingredients protect against neurological diseases [6, 7] and improve cognitive impairment in Aβ 1-42-induced neurodegeneration in mice [8].”
Nowak, A., et al., Potential of Schisandra chinensis (Turcz.) Baill. in Human Health and Nutrition: A Review of Current Knowledge and Therapeutic Perspectives. Nutrients, 2019. 11(2). Li, C.L., Y.H. Tsuang, and T.H. Tsai, Neuroprotective Effect of Schisandra Chinensis on Methyl-4-Phenyl-1,2,3,6-Tetrahydropyridine-Induced Parkinsonian Syndrome in C57BL/6 Mice. Nutrients, 2019. 11(7).
Q9) Methods: are adequately described, only a few minor points, as follows:
L.233. “and the supernatant was”, I think the authors wanted to mean the filtrate instead of supernatant
L.242-243: The authors should mention if the H2O2 treatment was performed in the presence of extracts or if the extracts incubation media was removed prior to challenge the cells with hydrogen peroxide.
Also the concentration of hydrogen peroxide should be mentioned (the range of concentrations).
L245-246. Please verify if there is a missing step.
A9) Thanks.
First, we modified “supernatant” to “filtrate”.
Second, to avoid confusion, we clearly mentioned in the result section that the control means 50 μM of H2O2 in Figure 2, Figure 3 and supplementary figure 1.
Third, we added the concentration of hydrogen peroxide in figure legends and figures.
Fourth, we described the EZ-Cytox cell viability assay in more detail as described in below, on line 23, page 14.
“We used the EZ-Cytox Cell Viability Assay Kit (Daeil, Korea) to assess cell viability. After serial treatment of SC, RF and hydrogen peroxide, WST (20 μl; 5 mg/ml in phosphate-buffered solution) was added to each well and the cells were incubated for another 4 h. WST is reduced by dehydrogenase in cells to form an orange‑colored formazan. The amount of formazan and the number of living cells are in direct proportion. Absorbance was measured at wavelength of 450 nm and 655 nm (for reference) using a microplate reader (Bio-Rad, Hercules, CA, USA).”
Q10) Supplementary material. Supplementary Table 1., should mention in its legend the type of extracts (% of ethanol).
A10) Thank you for your comment. In supplementary table, extracted all Korean plants with 30% EtOH. We mentioned in legend of supplementary table1.
References
박은주, T.A.P.E.J., J.-J. Ahn, and T.A.K.J.-H. 권중호, 열수추출조건이 동결건조 오미자의 추출 및 항산화 특성에 미치는 영향. 2013. 45. Giridharan, V.V., et al., Prevention of scopolamine-induced memory deficits by schisandrin B, an antioxidant lignan from Schisandra chinensis in mice. Free Radic Res, 2011. 45(8): p. 950-8. Song, J.X., et al., Protective effects of dibenzocyclooctadiene lignans from Schisandra chinensis against beta-amyloid and homocysteine neurotoxicity in PC12 cells. Phytother Res, 2011. 25(3): p. 435-43. Zhai, J., et al., Schisandra chinensis extract decreases chloroacetaldehyde production in rats and attenuates cyclophosphamide toxicity in liver, kidney and brain. J Ethnopharmacol, 2018. 210: p. 223-231. Nowak, A., et al., Potential of Schisandra chinensis (Turcz.) Baill. in Human Health and Nutrition: A Review of Current Knowledge and Therapeutic Perspectives. Nutrients, 2019. 11(2). Zhang, M., L. Xu, and H. Yang, Schisandra chinensis Fructus and Its Active Ingredients as Promising Resources for the Treatment of Neurological Diseases. Int J Mol Sci, 2018. 19(7). Li, C.L., Y.H. Tsuang, and T.H. Tsai, Neuroprotective Effect of Schisandra Chinensis on Methyl-4-Phenyl-1,2,3,6-Tetrahydropyridine-Induced Parkinsonian Syndrome in C57BL/6 Mice. Nutrients, 2019. 11(7). Zhao, X., et al., Total Lignans of Schisandra chinensis Ameliorates Abeta1-42-Induced Neurodegeneration with Cognitive Impairment in Mice and Primary Mouse Neuronal Cells. PLoS One, 2016. 11(4): p. e0152772.
Reviewer 2 Report
In the manuscript submitted to International Journal of Molecular Sciences (code 585391) authors works on the synergistic neuroprotective effect of schisandra chinensis and ribes fasciculatum on neuronal cell death and scopolamine-induced cognitive impairment in rats. This reviewer suggest the publication in IJMS after minor revision.
Theme is interesting, techniques are the adequated to solve this kind of determinations.
Minor comments:
* Introduction must be improved, providing more cites and information.
* Line 228: please provide as a cite the link (http://extract.kribb.re.kr).
* In Experimental, for Instrumentation, Materials and Reagents, or Programs and Databases (as SPSS, Excel, and others) ever, Product (Manufacturer, City, Country), in this order and format. Please correct in some places. In the case of USA products: Product (Manufacturer, City, State, USA).
* Conclusions must be improved.
* The Supplementary data are necessary.
Author Response
Comments and Suggestions for Authors
In the manuscript submitted to International Journal of Molecular Sciences (code 585391) authors works on the synergistic neuroprotective effect of schisandra chinensis and ribes fasciculatum on neuronal cell death and scopolamine-induced cognitive impairment in rats. This reviewer suggest the publication in IJMS after minor revision.
Theme is interesting, techniques are the adequated to solve this kind of determinations.
We deeply thank the reviewer. The Introduction and Conclusions have been rewritten to improve our findings.
Minor comments:
Q1) Introduction must be improved, providing more cites and information.
A1) Thank you for constructive comment. The Introduction was rewritten to support the rationale to our study.
“Mild cognitive impairment (MCI) is known as early dementia and isolated memory impairment, affecting elderly people [1]. Delays or prevention of MCI are important because people with MCI are at high risk for developing Alzheimer's or other dementia [2, 3]. However, United States Food and Drug Administration (FDA)-approved drugs inhibiting the cholinesterase or N-methyl-d-aspartate receptor, such as Aricept (donepezil), Exelon (rivastigmine), and Namenda (memantine), only mitigate the symptoms of Alzheimer’s disease[4-6]. Moreover, some of these drugs should not be used in the long term due to affect the heart rate as well as the nervous system, causing side effects such as arrhythmia [7]. To overcome the drawback of modern therapeutics, Natural plants have emerged as alternative medicine for prevention therapy for transitional state of disease which have low risk of developing side effects [8].
Schisandra chinensis (SC), a deciduous woody vine, has been used as a traditional herbal medicine for the treatment of cancer and diabetes via immunomodulatory and antioxidant activities [9]. Bioactive compounds such as schisandrin B, a dibenzocyclooctadiene derivative isolated from SC, have been studied for their neuroprotective effects in scopolamine-induced amnesia, amyloid beta (Aβ) peptide-induced neurotoxicity, and cisplatin-induced neurotoxicity [10, 11]. In addition, schisanhenol isolated from SC has been shown to improve learning and memory in scopolamine-treated mice [12]{Han, 2018 #3;Han, 2019 #24}. Although protective effects of various extracts of SC were reported on neuronal injury and memory impairment, but the underlying mechanism of it still did not defined with BDNF signaling, and revealed that correlation with antioxidant enzyme activation [13, 14], NF-kB inhibition [15] and recovered Ach levels [10, 16].
Here, we found out protective role of neuronal cell death by Ribes fasciculatum (RF), a perennial woody deciduous member of the Ribes genus in the family Grossulariaceae, is distributed widely in Korea and China. Previous studies have reported that RF has been shown to increase the lifespan and stress resistance of Caenorhabditis elegans through SIR-2.1-mediated DAF-16 activation depending on the insulin/insulin growth factor signaling pathway [17]. Moreover, RF inhibited the transcription of nuclear factor of activated T cells (NFAT) [18] and the secretion of inflammatory cytokines in a mouse model of atopic dermatitis [19]. Since nuclear factors of activated T cells (NFATc4) have been reported to be required for BDNF-dependent survival of adult-born neurons and spatial memory formation in the hippocampus, we focused on the possibility that combination of SC and RF could enhance protection against cognitive impairment through the BDNF signaling pathway.
Ultimately, in the present study, we aimed to investigate whether SC and RF extracts have a protective effect on reactive oxygen species (ROS)-induced cytotoxicity in pheochromocytoma PC12 cells and scopolamine-induced cognitive impairment in rats.”
Q2) Line 228: please provide as a cite the link (http://extract.kribb.re.kr).
A2) Thank you for point out. We provide a cite the link and reference.
“A total of 42 plant extracts were originally provided from the Korea Plant Extract Bank at Korea Research Institute of Bioscience & Biotechnology (KRIBB) (http://extract.kribb.re.kr, Ochang, Chungbuk, Korea)[20].”
Ahn, K., The worldwide trend of using botanical drugs and strategies for developing global drugs. BMB Rep, 2017. 50(3): p. 111-116.
Q3) In Experimental, for Instrumentation, Materials and Reagents, or Programs and Databases (as SPSS, Excel, and others) ever, Product (Manufacturer, City, Country), in this order and format. Please correct in some places. In the case of USA products: Product (Manufacturer, City, State, USA).
A3) Thanks. We correct the information in the Method section.
Q4) Conclusions must be improved.
A4) We have improved the conclusion as shown below.
“Our study showed that SC and RF protected neuronal cells against H2O2-induced cell death. Moreover, SC and RF prevented cognitive impairment via BDNF signaling pathway in scopolamine induced rat model. Our findings indicate the potential of SC and RF for use in the prevention and treatment of cognitive impairment in humans with Alzheimer’s disease.”
Q5) The Supplementary data are necessary.
A5) We attached the supplementary data. Please find it below. Thank you.
Supplementary Figures
Synergistic neuroprotective effect of Schisandra chinensis and Ribes fasciculatum on neuronal cell death and scopolamine-induced cognitive impairment in rats
Eunkuk Park, Min Jeong Ryu, Nam Ki Kim, Mun Hyoung Bae, Youngha Seo, Jeonghyun Kim, , Subin Yeo, Memoona Kanwal, Chun Whan Choi, Jun Young Heo and Seon-Yong Jeong
Figure S1. The effect of different ethanol extracts of SC and RF on cell survival rate. SC and RF were extracted with four different concentrations of ethanol (0%, 30%, 70%, or 100%) and PC12 neuronal cells were treated with 2, 10, or 50 mg/ml of SC or RF extracts. The protective effect of the extracts was measured by the WST assay. Control is a treatment of 50 µM of H2O2. The data shown are means ± SEM. *: p < 0.05 vs. control.
Figure S2. SC and RF did not affect the proliferation of neuronal cells. The cells were cultured with SC or RF (10 or 50 μg/ml), and cell viability was analyzed.
Figure S3. A combination of SC and RF did not affect PC12 cell proliferation. The cells were treated with different ratios (6:4, 7:3, or 8:2) of a combination of SC and RF (10 or 50 μg/ml), and cell viability was analyzed.
Figure S4. HPLC profiles of the SC (A) and RF (B) and combination of SC and RF (7:3) (C) extracts obtained by HPLC-ESI-MS analysis Chemical structures (D) of the main constituent of shizandrin ⓐ identified in SC and 4-hydrobenzoic acid ⓑ identified in RF.
Figure S5. Changes in escape latency time over 5 days in scopolamine-treated rats. Seven-week-old male Sprague Dawley® rats were administered phosphatidylserine orally and a mixture of SR (SC+RF) for 23 days before scopolamine injection. Escape latency time was measured daily for 5 days. Abbreviation: PS, phosphatidylserine.
Figure S6. Total body weight changes during the passive avoidance test (A) and the Morris water maze (B) experiments. Rats were treated orally with SR (SC+RF) (75, 150 or 300 mg/kg/day) for 23 days and memory impairment was measured by the Morris water maze and the passive avoidance test for five days. Total body weight was measured in each week. Abbreviation: PS, phosphatidylserine.
References
Knopman, D.S. and R.C. Petersen, Mild cognitive impairment and mild dementia: a clinical perspective. Mayo Clin Proc, 2014. 89(10): p. 1452-9. Andrade, C. and R. Radhakrishnan, The prevention and treatment of cognitive decline and dementia: An overview of recent research on experimental treatments. Indian J Psychiatry, 2009. 51(1): p. 12-25. Kantarci, K., et al., Risk of dementia in MCI: combined effect of cerebrovascular disease, volumetric MRI, and 1H MRS. Neurology, 2009. 72(17): p. 1519-25. Cacabelos, R., Donepezil in Alzheimer's disease: From conventional trials to pharmacogenetics. Neuropsychiatr Dis Treat, 2007. 3(3): p. 303-33. Onor, M.L., M. Trevisiol, and E. Aguglia, Rivastigmine in the treatment of Alzheimer's disease: an update. Clin Interv Aging, 2007. 2(1): p. 17-32. van Marum, R.J., Update on the use of memantine in Alzheimer's disease. Neuropsychiatr Dis Treat, 2009. 5: p. 237-47. Masuda, Y., Cardiac effect of cholinesterase inhibitors used in Alzheimer's disease--from basic research to bedside. Curr Alzheimer Res, 2004. 1(4): p. 315-21. Atanasov, A.G., et al., Discovery and resupply of pharmacologically active plant-derived natural products: A review. Biotechnol Adv, 2015. 33(8): p. 1582-1614. Li, Z., et al., A review of polysaccharides from Schisandra chinensis and Schisandra sphenanthera: Properties, functions and applications. Carbohydr Polym, 2018. 184: p. 178-190. Giridharan, V.V., et al., Prevention of scopolamine-induced memory deficits by schisandrin B, an antioxidant lignan from Schisandra chinensis in mice. Free Radic Res, 2011. 45(8): p. 950-8. Song, J.X., et al., Protective effects of dibenzocyclooctadiene lignans from Schisandra chinensis against beta-amyloid and homocysteine neurotoxicity in PC12 cells. Phytother Res, 2011. 25(3): p. 435-43. Han, Y., et al., Schisanhenol improves learning and memory in scopolamine-treated mice by reducing acetylcholinesterase activity and attenuating oxidative damage through SIRT1-PGC-1alpha-Tau signaling pathway. Int J Neurosci, 2019. 129(2): p. 110-118. Hu, D., et al., Deoxyschizandrin isolated from the fruits of Schisandra chinensis ameliorates Abeta(1)(-)(4)(2)-induced memory impairment in mice. Planta Med, 2012. 78(12): p. 1332-6. Hu, D., et al., Schizandrin, an antioxidant lignan from Schisandra chinensis, ameliorates Abeta1-42-induced memory impairment in mice. Oxid Med Cell Longev, 2012. 2012: p. 721721. Giridharan, V.V., et al., Schisandrin B, attenuates cisplatin-induced oxidative stress, genotoxicity and neurotoxicity through modulating NF-kappaB pathway in mice. Free Radic Res, 2012. 46(1): p. 50-60. Wei, B.B., et al., Schisandrin ameliorates cognitive impairment and attenuates Abeta deposition in APP/PS1 transgenic mice: involvement of adjusting neurotransmitters and their metabolite changes in the brain. Acta Pharmacol Sin, 2018. 39(4): p. 616-625. Jeon, H. and D.S. Cha, Anti-aging properties of Ribes fasciculatum in Caenorhabditis elegans. Chin J Nat Med, 2016. 14(5): p. 335-42. Dat, N.T., et al., New inhibitor against nuclear factor of activated T cells transcription from Ribes fasciculatum var. chinense. Chem Pharm Bull (Tokyo), 2005. 53(1): p. 114-7. Jung, J.W., et al., Ribes fasciculatum var. chinense Attenuated Allergic Inflammation In Vivo and In Vitro. Biomol Ther (Seoul), 2014. 22(6): p. 547-52. Ahn, K., The worldwide trend of using botanical drugs and strategies for developing global drugs. BMB Rep, 2017. 50(3): p. 111-116.
Round 2
Reviewer 1 Report
The authors rigorously made the necessary corrections to the manuscript, substantially improving it. now meets conditions to be published